# Contrastive Pre-training for Personalized Expert Finding

**Qiyao Peng[1], Hongtao Liu[2]\*, Zhepeng Lv[2], Qing Yang[2], Wenjun Wang[1,3,4]†**
[1] School of New Media Communication, Tianjin University, Tianjin, China
[2] Du Xiaoman Financial, Beijing, China
[3] Georgia Tech Shenzhen Institute, Tianjin University, Guangdong, China
[4] Hainan Tropical Ocean University, Hainan, China
[1]{qypeng, wjwang}@tju.edu.cn
[2]{liuhongtao01, lvzhepeng, yangqing}@duxiaoman.com

## Abstract

Expert finding could help route questions to potential suitable users to answer in Community Question Answering (CQA) platforms. Hence it is essential to learn accurate representations of experts and questions according to the question text articles. Recently the pre-training and fine-tuning paradigms are powerful for natural language understanding, which has the potential for better question modeling and expert finding. Inspired by this, we propose a CQA-domain Contrastive Pre-training framework for Expert Finding, named CPEF, which could learn more comprehensive question representations. Specifically, considering that there is semantic complementation between question titles and bodies, during the domain pre-training phase, we propose a title-body contrastive learning task to enhance question representations, which directly treats the question title and the corresponding body as positive samples of each other, instead of designing extra data-augmentation strategies. Furthermore, a personalized tuning network is proposed to inject the personalized preferences of different experts during the fine-tuning phase. Extensive experimental results on six real-world datasets demonstrate that our method could achieve superior performance for expert finding.

## 1 INTRODUCTION

Nowadays, Community Question Answering (CQA) platforms such as StackExchange [1], BaiduZhidao [2] have been popular and extremely attracted millions of users (Fu et al., 2020), which can raise their questions or post answers for questions they are interested in or good at. Due to the large participation, there are too many questions to wait for answers (Zhao et al., 2017; Yuan et al.,

2020). In order to alleviate that, it needs to recommend suitable experts to provide satisfactory answers (Chang and Pal, 2013; Zhao et al., 2014) for these questions. Expert finding aims to effectively route questions to a suitable expert based on her/his historically answered questions, which has attracted considerable attention recently.

Based on the powerful neural network (LeCun et al., 2015), many approaches (Li et al., 2019; Fu et al., 2020; Peng et al., 2022a) have been shown effective to improve the performance of expert finding. The core idea is to first model the question semantically, and then learn the expert preference based on his/her historically answered questions. For example, Peng et al. (Peng et al., 2022a) proposed PMEF equipped with a multi-view question modeling paradigm aiming to model questions more comprehensively and then capture expert features. Hence, the capacity of the designed question modeling paradigm directly affects the expert finding performance.

Recently, Pre-trained Language Models (PLMs) have achieved great success in Natural Language Processing (NLP) due to their strong ability in text modeling (Devlin et al., 2018). Different from traditional models that are usually directly trained with labeled data in specific tasks, PLMs are usually pre-trained on a large unlabeled corpus via self-supervision to encode universal semantic information. However, existing PLMs are usually pre-trained on general corpus such as BookCorpus and Wikipedia, which have some gaps with the CQA domain and the expert finding task. Directly fine-tuning the existing PLMs with the expert finding task may be sub-optimal for CQA question understanding. Besides, such an architecture does not take into consideration the expert's personalized characteristics, which could impact expert modeling and question-expert matching (i.e., expert finding). At present, **the exploration of the pre-training paradigm in expert finding is pre-**

---

Hongtao Liu is the co-first author.
†Wenjun Wang is the corresponding author.

[1]https://stackexchange.com
[2]https://zhidao.baidu.com

**liminary**.

In this paper, we aim to design a novel expert finding framework, which could learn more comprehensive question and expert representations with domain pre-training and personalized fine-tuning paradigm. For pre-training question representations, in addition to the domain pre-training Masked Language Model (MLM) task, we design a contrastive task to capture the inherent semantic information from the unlabeled CQA question corpus. Different from the vanilla contrastive learning paradigm with data augmentation progress, we directly consider the question title and the question body as mutually augmented perspectives. In this way, the model could omit the tedious process of data augmentation while incorporating the two relevant types of information (i.e., question title and body) for question semantic modeling. For fine-tuning phase, we transfer the pre-trained question representations to model expert representations and the downstream expert finding task. Considering different experts would have different tastes in the same question, hence we design a personalized tuning network, which could effectively inject personalized preference (i.e., expert ID) to learn more personalized expert representations for better question-expert matching.

In summary, the contributions of our method are:

- We propose a domain pre-training and fine-tuning framework for expert finding on CQA platforms, which could learn comprehensive representations of questions and experts.

- We design a novel question title-body contrastive pre-training task, which could improve question modeling effectively. Further, we adopt a personalized tuning network to learn personalized representations of experts.

- We conduct extensive experiments on six real-world CQA datasets. Experimental results show that our method can achieve better performance than existing baselines and validate the effectiveness of our approach.

## 2 RELATED WORK

In this section, we briefly review the related work in two aspects, namely Expert Finding and Pre-training for Recommendation.

Expert finding has received much attention from both research and industry community (Liu et al.,

2015; Yuan et al., 2020; Liu et al., 2022; Peng and Liu, 2022; Peng et al., 2022b) and it could help find capable experts for providing answers to questions. Early studies employed feature-engineering (Cao et al., 2012; Pal et al., 2012) or topic-modeling technology (Guo et al., 2008; Riahi et al., 2012) to model questions and experts, then measured similarities between them for routing questions to suitable experts. Subsequently, with the development of deep learning (LeCun et al., 2015), recent methods employed neural networks to learn expert and question representations, then matched them (Li et al., 2019; Fu et al., 2020; Ghasemi et al., 2021). For example, Peng et al. (Peng et al., 2022a) designed a multi-view learning paradigm to learning question features, which could improve expert finding.

Different from most existing methods learning question representations that rely on supervised data, we design a question title-body contrastive pre-training paradigm for modeling question. In this way, the model could capture more comprehensive question semantic representations, which are the basis of expert modeling and question-expert matching.

Furthermore, expert finding aims to recommend suitable experts for answering target questions, which is similar to recommender system. Hence, we will introduce some recommendation researches with pre-training (Zhou et al., 2020; Wu et al., 2020; Cheng et al., 2021; Hou et al., 2022) in this part. For example, Cheng et al. (Cheng et al., 2021) proposed CLUE, which designed contrastive pre-training tasks to optimize transferable sequence-level user representations for recommendation. Compared with existing recommendation works under the pre-training paradigm, the expert finding has different characteristics, e.g., the target question is usually cold-started and ID-based recommendation may not be feasible.

It is noted that the very recent method ExpertBert (Liu et al., 2022) designs a preliminary expert-specific pre-training framework, which has achieved the best performance. Different from that, our approach focuses on more comprehensive question semantic representations via integrating the title-body contrastive learning task during pre-training. In addition, we design a personalized tuning network to tune the model and model expert according to different expert preferences.

## 3 Problem definition

Expert finding in CQA websites is to predict the most suitable expert to answer questions. Suppose that there is a target question $q^t$ and a candidate expert set $E = \{e_1, \cdots, e_M\}$ respectively, where $M$ is the number of experts. Given a candidate expert $e_i \in E$, she/he is associated with a set of her/his historical answered questions, which can be denoted as $Q_i = \{q_1, \cdots, q_n\}$ where $n$ is the number of historical questions. In addition to the title, each question contains the body information, which is a detailed description of the question; however, most existing methods ignore the question bodies. In fact, the question body could be a complementation of the question title. Hence, in this paper we would explore how to utilize the title and body together effectively.

## 4 Proposed Method

In this section, we will introduce the training procedure of our method *CPEF*, which is composed of the following stages.

- In pre-training phase (question modelling), based on the CQA domain unlabeled text corpus, we propose a contrastive learning task between the question title and body and integrate it into the pre-training Masked Language Model (MLM) task jointly, to fully learn question semantic representations.

- In fine-tuning phase (expert finding), we initialize the parameters of the question representation model with the pre-trained parameters and then utilize a personalized tuning network to fine-tune the model with traditional expert finding supervised signals.

### 4.1 Pre-training Framework

In this part, we will present the question representation pre-training framework, including the well-designed input layer and a brief backbone architecture introduction.

**(1) Input Layer** For the model input, we concatenate the words of the question title and the question body into a whole sequence as shown in Figure 1. We propose to design special tokens ([TITLE] and [BODY]) at the beginning of the title sequence and the body sequence respectively, to indicate the title text and the body text. Moreover, the special

token [SEP] is added at the end of the input word sequence for recognizing the end of the sequence. Hence, the question information is denoted as:

$$\mathcal{Q} = [[\text{TITLE}], \underbrace{[tk_1], [tk_2], \cdots}_{title}, [\text{BODY}],$$
$$\underbrace{[tk_1], [tk_2], [tk_3], \cdots}_{body}, [\text{SEP}]] \quad (1)$$

Then, the input representation matrix $\mathbf{R}$ is constructed by summing its corresponding token, segment, and position embedding:

$$\mathbf{R} = \mathbf{R}_{token} + \mathbf{R}_{seg} + \mathbf{R}_{pos}, \quad (2)$$

where $\mathbf{R}_{token}$ is the embedding matrix of tokens derived from tokenizing $\mathcal{Q}$ (with BERT Tokenizer), and $\mathbf{R}_{seg}$ and $\mathbf{R}_{pos}$ are corresponding segment embedding and position embedding respectively following the settings in BERT (Devlin et al., 2018). It is noted that the question title segment ID is 0 and the question body segment id is 1, which could help distinguish between these two types of information.

**(2) Model Architecture** In our method, we adopt the widely used BERT as our base backbone encoder to further pre-train question representations in the CQA domain corpus. BERT architecture consists of multi-layer Transformer encoder layers. Each Transformer encoder layer has the following two major sub-layers, i.e., multi-head self-attention and position-wise feed-forward layer.

*Multi-Head Self-Attention.* This sub-layer aims to capture the contextual representations for each word. The self-attention function is defined as:

$$\text{Att}(\mathbf{Q}, \mathbf{K}, \mathbf{V}) = \text{Softmax}(\mathbf{Q}\mathbf{K}^T/\sqrt{d})\mathbf{V}, \quad (3)$$

where $\mathbf{Q}$, $\mathbf{K}$ and $\mathbf{V}$ represent the query, key and value matrix correspondingly. Multi-head self-attention layer $MH(\cdot)$ will project the input to multiple sub-spaces and capture the interaction information, which is denoted as:

$$\text{MH}(\mathbf{R}) = [head_1; \cdots; head_h]\mathbf{W}, \quad (4)$$

$$head_i = \text{Att}(\mathbf{X}\mathbf{W}_i^q, \mathbf{X}\mathbf{W}_i^k, \mathbf{R}\mathbf{W}_i^v), \quad (5)$$

where $\mathbf{W}_i^q, \mathbf{W}_i^k, \mathbf{W}_i^v \in \mathcal{R}^{d \times \frac{d}{h}}$ and $\mathbf{W} \in \mathcal{R}^{d \times d}$ are parameters. Via the multi-head self-attention, the representation matrix $\mathbf{R}$ is transformed to $\mathbf{H} \in \mathcal{R}^{n \times d}$, where $n$ is the token number.

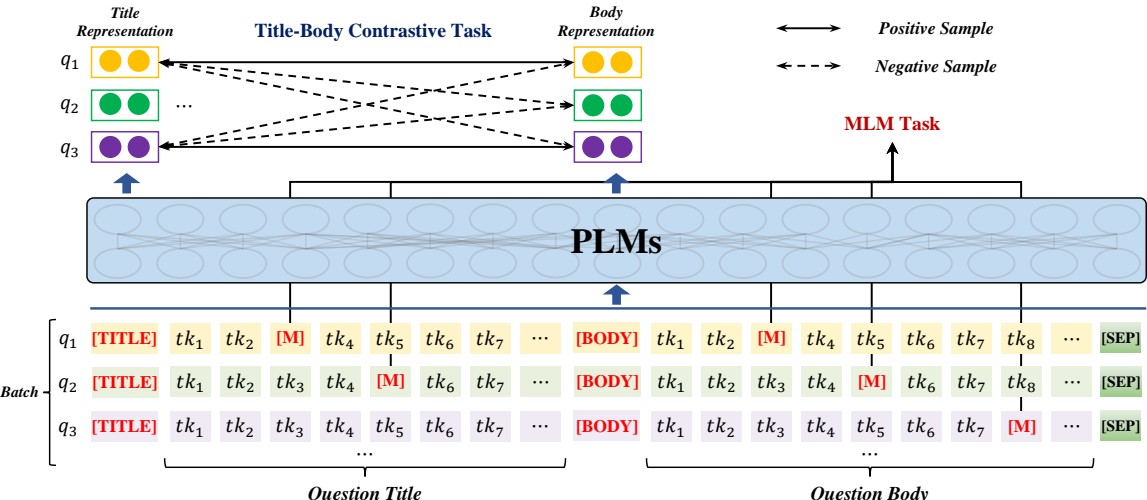

Figure 1: CPEF Question Modelling: **Contrastive Pre-training Framework**. We employ the title-body contrastive task and masked language model task to jointly learn more comprehensive question representations. $tk$ denotes the token and $q$ represents the question.

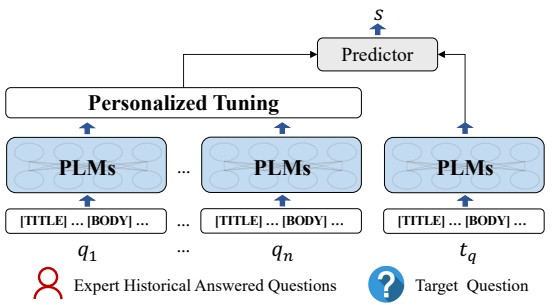

Figure 2: CPEF Expert Modelling: **Personalized Fine-tuning Network**.

*Position-wise feed-forward.* For the input $\mathbf{H}$, the calculation is defined as:

$$\text{FFN}(\mathbf{H}) = \text{RELU}(\mathbf{H}\mathbf{W}_1^f + b_1^f)\mathbf{W}_2^f + b_2^f, \quad (6)$$

where $\mathbf{W}_1^f$, $\mathbf{W}_2^f$ and $b_1^f$, $b_2^f$ are learnable parameters. More details please refer to the BERT (Devlin et al., 2018).

Then, we will next introduce our well-design pre-training tasks, which could help capture more comprehensive question representations.

### 4.2 Pre-training task

Having established the input information, during the pre-training stage, we propose to employ two self-supervised tasks (including the contrastive task and mask language model task) for question pre-training.

**Contrastive Task.** This task is to discriminate the representation of a particular question from other questions. Considering the question title and the question body could describe the question from two different perspectives, we directly consider the question title and the question body as positive samples mutually. In this way, the model could omit the tedious process of data augmentation and combine two types of information (i.e., question title and question body) by contrastive learning.

As shown in Figure 1, given the $i$-th question, its title representation $\mathbf{q}_i^t$ and its body representation $\mathbf{q}_i^b$ are positive samples of each other, while other **in-batch questions** are considered as negatives. The contrastive task can be denoted as:

$$\mathcal{L}_q = -\sum_{i=1}^{B} \log \frac{\exp(\text{sim}(\mathbf{q}_i^t \cdot \mathbf{q}_i^b)/\tau)}{\sum_{i'=1}^{B} \exp(\text{sim}(\mathbf{q}_i^t \cdot \mathbf{q}_{i'}^b)/\tau)},$$
$$(7)$$

where $\mathbf{q}_i^t$ is $i$-th question title representation and $\mathbf{q}_{i'}^b$ is the question body representation of the negative instance. $\tau$ is the temperature. Through this contrastive learning module, our model could learn more precise and distinguished representations of questions.

Furthermore, for helping capture the CQA language knowledge, inspired by BERT (Devlin et al., 2018), we simply mask some percentage of the input tokens randomly and then predict those masked tokens. The formal definition is:

$$\mathcal{L}_m = -\sum_{i=1}^{\mathcal{N}} \log p(tk = tk_i|\theta), tk_i \in [1, 2, \cdots, \mathcal{V}]$$
$$(8)$$

where $\mathcal{N}$ is the masked token set, $\mathcal{V}$ is the vocabulary set.

**Model Pre-training.** At the pre-training stage, we leverage a multi-task training strategy to jointly pre-train the model via the contrastive learning task and the MLM task, denoted as:

$$\mathcal{L}_{pt} = \mathcal{L}_q + \mathcal{L}_m . \qquad (9)$$

### 4.3 Personalized fine-tuning

We have obtained the pre-trained question representation model based on the unlabeled CQA corpus. In this section, we will tune the pre-trained model for evaluating the expert finding task. For realizing the task, we need to model experts based on their historical answered questions.

Despite modeling questions more comprehensively during the pre-training stage, such an architecture is less capable in modeling personalized preferences for experts. Intuitively, an expert is likely to show personalized tastes when answering questions. Hence, our solution is to design an expert-level personalized tuning mechanism for injecting personalized preference while modeling experts.

Specifically, as illustrated in Figure 2, given the $i$-th expert historical answered questions (i.e., $Q_i = \{q_1, \cdots, q_n\}$), we first employ the pre-trained question modeling weight to obtain each historical question representations ($\mathbf{q}_1, \cdots, \mathbf{q}_n$). Then, we design a personalized tuning mechanism to compute the attention weight $\alpha_j$ of $j$-th historical answered question with respect to the expert, which is denoted as follows:

$$\alpha_j = \frac{\exp(l_j)}{\sum_{k=1}^{n} \exp(l_k)}, \; l_j = (\mathbf{o}_i)^T \odot \mathbf{q}_j, \quad (10)$$

where $\mathbf{o}_i$ and $\mathbf{q}_j$ are representations of the $i$-th expert ID embedding and the $j$-th historical answered question respectively. In this way, the model could capture the expert's personalized interests on different questions.

Finally, for obtaining the expert representation, we aggregate historical question features according to their weights:

$$\mathbf{e}_i = \sum_{j=1}^{n} \alpha_j \mathbf{q}_j, \qquad (11)$$

where $\mathbf{e}_i$ is the expert $i$ final representation.

For fine-tuning the model, we use the negative sampling technology (Huang et al., 2013) to sample $K$ negatives and the cross-entropy loss, which is

| Datasets | # questions | # answerers | # answers |
|----------|-------------|-------------|-----------|
| Es | 68,104 | 9,539 | 94,393 |
| Tex | 129,202 | 6,999 | 189,368 |
| Unix | 96,258 | 15,581 | 163,553 |
| English | 51,498 | 10,655 | 120,414 |
| Physics | 85,776 | 10,117 | 143,570 |
| Electronics | 78,954 | 7,402 | 143,522 |

Table 1: Statistical details of the datasets.

denoted as:

$$s = \frac{\exp(s)}{\sum_{j=1}^{K+1} \exp(s_j)} , \mathcal{L}_{ft} = -\sum_{c=1}^{K+1} \hat{s} \log(s) , \qquad (12)$$

where $\hat{s}$ is the ground truth label and $s$ is the normalized probability predicted by the model.

## 5 Experiments

In this section, we first introduce the experiment settings and then present the results and analysis.

### 5.1 Datasets and Experimental Settings

For pre-training question representations, we construct a dataset containing 525,519 unlabeled question data (including question title and question body), which is from StackExchange[3]. For fine-tuning and verifying the effect of the pre-trained model in specific domains, we select six different domains, i.e., **Es**, **Tex**, **Unix**, **Physics**, **English** and **Electronics**. Each dataset includes a question set, in which, each question is associated with its title, body, and an "accepted answer" among several answers provided by different answerers. The detailed statistical characteristics of the datasets are shown in Table 1.

We split each dataset into a training set, a validation set and a testing set, with the ratios 80%, 10%, 10% respectively in chronological order. Specifically, we filter out experts who did not answer twice for mitigating the cold start problem. We pad or truncate the length of the question title and body as 10 and 30 respectively. For pre-training, we set the ratio of masking tokens is 0.2 and the temperature $\tau$ is 0.1. For fine-tuning, following the previous works, we set the number of negative samples $K$ is 19. For testing methods, we randomly assigned 80 experts to each question (including those who had answered the question) and the baseline methods were reproduced as such.

[3]https://archive.org/details/stackexchange

| Dateset | Physics | | | Tex | | | English | | |
|---|---|---|---|---|---|---|---|---|---|
| Metric / Method | MRR | P@3 | NDCG@10 | MRR | P@3 | NDCG@10 | MRR | P@3 | NDCG@10 |
| CNTN | 0.3088 | 0.3837 | 0.4036 | 0.3825 | 0.4513 | 0.4612 | 0.3015 | 0.3001 | 0.3653 |
| NeRank | 0.4801 | 0.5734 | 0.5608 | 0.5891 | 0.6734 | 0.6632 | 0.4086 | 0.4633 | 0.5089 |
| TCQR | 0.4025 | 0.4937 | 0.5087 | 0.4733 | 0.5553 | 0.5634 | 0.3525 | 0.3621 | 0.4053 |
| RMRN | 0.4720 | 0.5676 | 0.5522 | 0.5890 | 0.6786 | 0.6655 | 0.4023 | 0.4611 | 0.5099 |
| UserEmb | 0.3773 | 0.4656 | 0.4536 | 0.4256 | 0.5223 | 0.5333 | 0.3277 | 0.3362 | 0.3915 |
| PMEF | 0.4835 | 0.5723 | 0.5615 | 0.5966 | 0.7018 | 0.6711 | 0.4099 | 0.4732 | 0.5120 |
| ExpertBert | 0.4908 | 0.5803 | 0.5635 | 0.6042 | 0.7033 | 0.6735 | 0.4126 | 0.4812 | 0.5155 |
| **CPEF** | **0.5051** | **0.5909** | **0.5753** | **0.6209** | **0.7204** | **0.6896** | **0.4276** | **0.4936** | **0.5211** |
| Dateset | Electronics | | | Unix | | | Es | | |
| Metric / Method | MRR | P@3 | NDCG@10 | MRR | P@3 | NDCG@10 | MRR | P@3 | NDCG@10 |
| CNTN | 0.3188 | 0.3537 | 0.3936 | 0.3529 | 0.4313 | 0.4416 | 0.2998 | 0.3121 | 0.3358 |
| NeRank | 0.5111 | 0.6012 | 0.6008 | 0.5628 | 0.6116 | 0.6423 | 0.4813 | 0.5386 | 0.5369 |
| TCQR | 0.4075 | 0.5037 | 0.5127 | 0.4836 | 0.5353 | 0.5734 | 0.3825 | 0.4621 | 0.4453 |
| RMRN | 0.5232 | 0.6111 | 0.6172 | 0.5723 | 0.6232 | 0.6521 | 0.4912 | 0.5393 | 0.5399 |
| UserEmb | 0.3623 | 0.4758 | 0.4699 | 0.4156 | 0.4823 | 0.5133 | 0.3077 | 0.3662 | 0.3715 |
| PMEF | 0.5423 | 0.6302 | 0.6282 | 0.5888 | 0.6412 | 0.6687 | 0.5093 | 0.5588 | 0.5501 |
| ExpertBert | 0.5433 | 0.6301 | 0.6199 | 0.5988 | 0.6521 | 0.6725 | 0.5052 | 0.5601 | 0.5575 |
| **CPEF** | **0.5579** | **0.6426** | **0.6305** | **0.6064** | **0.6711** | **0.6799** | **0.5192** | **0.5726** | **0.5735** |

Table 2: Expert finding results of different methods. The best performance of the baselines is underlined. We perform t-test and the results show that **CPEF** outperforms other baselines at significance level p-value<0.05.

We adopt the pre-trained weight **bert-base-uncased** as the base model including 110M parameters. To alleviate the over-fitting problem, we utilize dropout technology (Srivastava et al., 2014) and set the dropout ratio as 0.2. We adopt Adam (Kingma and Ba, 2015) optimization strategy to optimize our model and set the learning rate to 5e-5 in further pre-training and 5e-2 in fine-tuning. The batch of pre-training and fine-tuning is 8. We employ pytorch, transformers, sklearn, numpy, accelerator, etc to implement our code. We independently repeat each experiment 5 times and report the average results. All experiments use two 24GB-memory RTX 3090 GPU servers with Intel(R) Xeon(R)@2.20GHz CPU.

## 5.2 Baselines and Evaluation metrics

We compare our method **CPEF** with recent competitive methods including:

- **CNTN** (Qiu and Huang, 2015) employs a CNN to model questions and computes relevance scores between questions and experts.

- **NeRank** (Li et al., 2019) learns question, raiser and expert representations via a HIN embedding algorithm and utilizes the CNN to match them.

- **TCQR** (Zhang et al., 2020) utilizes a temporal context-aware model in multiple temporal granularities to learn the temporal-aware expert representations.

- **RMRN** (Fu et al., 2020) employs a recurrent memory reasoning network to explore implicit relevance between expert and question.

- **UserEmb** (Ghasemi et al., 2021) utilizes a node2vec to capture social features and uses a word2vec to capture semantic features, then integrates them to improve expert finding.

- **PMEF** (Peng et al., 2022a) designs a personalized expert finding method under a multi-view paradigm, which could comprehensively model expert and question.

- **ExpertBert** (Liu et al., 2022) designs a expert-specific pre-training framework, towards precisely modelling experts based on the historical answered questions.

The evaluation metrics include Mean Reciprocal Rank (MRR) (Craswell, 2009), P@3 (i.e., Precision@3) and Normalized Discounted Cumulative Gain (NDCG@10) (Järvelin and Kekäläinen, 2002) to verify the expert ranking quality.

| Dateset | Electronics | | | Unix | | | Es | | |
|---|---|---|---|---|---|---|---|---|---|
| Metric
Method | MRR | P@3 | NDCG@10 | MRR | P@3 | NDCG@10 | MRR | P@3 | NDCG@10 |
| w/o CT | 0.5351 | 0.6111 | 0.6127 | 0.5722 | 0.6333 | 0.6339 | 0.4795 | 0.5423 | 0.5476 |
| w/o Per | 0.5401 | 0.6358 | 0.6266 | 0.5823 | 0.6576 | 0.6518 | 0.5005 | 0.5611 | 0.5628 |
| Original BERT | 0.5228 | 0.6016 | 0.5922 | 0.5619 | 0.6233 | 0.6176 | 0.4636 | 0.5328 | 0.5389 |
| **CPEF** | **0.5579** | **0.6426** | **0.6305** | **0.6064** | **0.6711** | **0.6799** | **0.5192** | **0.5726** | **0.5735** |

Table 3: The variants of CPEF experiment results.

## 5.3 Performance Comparison

We compare the proposed CPEF with the baselines on the six datasets. The experimental results of CPEF and other comparative methods are in Table 2. There are some observations from these results.

Some earlier methods (e.g., CNTN) obtain poorer results on almost datasets, the reason may be that they usually employ max or mean operation on histories to model expert, which omits different history importance. PMEF utilizes a multi-view question encoder to construct multi-view question features, which could help learn more comprehensive question representations, which is superior to other baseline methods.

Furthermore, the PLMs-based methods (ExpertBert, CPEF) achieve better performance. The reason is that the PLMs encoder could learn more precise semantic features of questions and experts compared with the traditional methods. Finally, compared with all baselines, our approach CPEF performs consistently better than them on six datasets. Different from the ExpertBert which only conducts MLM pre-training, we make full use of question titles and question bodies and design an extra title-body contrastive pre-training task, which allows the model to learn question representations more comprehensively in a self-supervised way. Furthermore, we utilize the personalized tuning mechanism to transfer and tune the pre-trained model according to different experts' personalized preferences. This result shows that our approach is effective to improve the performance of the expert finding.

## 5.4 Ablation Study

To highlight the effectiveness of our designing expert finding architecture, we design three model variants: **(a) w/o CT**, only adopt the corpus-level MLM to pre-train over CQA corpus and remove the title-body contrastive learning, and then per-

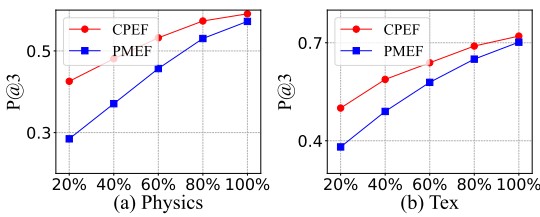

Figure 3: Impacts of Train Data Ratio.

sonalized fine-tune the model; **(b) w/o Per**, adopt the title-body contrastive learning and the MLM for pre-training but remove the personalized tuning mechanism during modeling experts; **(c) Original BERT**, directly adopt the original BERT weight for expert finding via a fine-tuning way, without pre-training and personalized tuning.

As shown in Table 3, we can have the following observations: (1) Regardless of removing the comparative learning in the pre-training (i.e., w/o CT) or the personalized tuning network in the fine-tuning (i.e., w/o Per), the model performance will decline. (2) **w/o Per** outperforms **w/o CT**. The reason is that the pre-trained question representations are the basis for subsequent expert modeling and question-expert matching. And the results show that the importance of the contrastive learning task in pre-training is greater than that of the personalized tuning. (3) It is not surprised that **BERT** obtains the worst performance. This is because the BERT is pre-trained on the general corpus, which causes the semantic gap with the CQA. Furthermore, directly fine-tuning BERT could not take the personalized characteristics into account. (4) Our complete model **CPEF** obtains the best results. The reason is it can pre-train more comprehensive question representations. Further, the personalized tuning network could capture expert personalized preference, which could yield better performance.

In all, the results of ablation studies meet our motivation and validate the effectiveness of our proposed method.

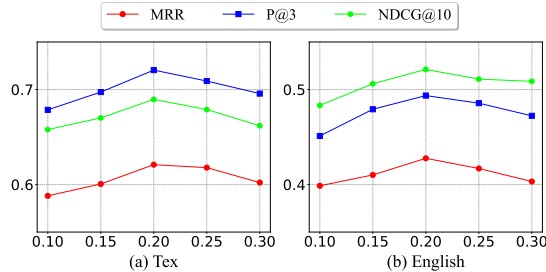

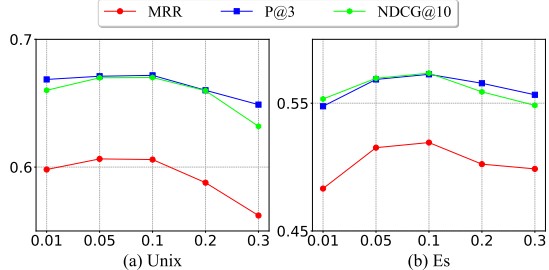

Figure 4: Impacts of different mask ratio.

Figure 5: Model performance w.r.t. $\tau$.

## 5.5 Further Analysis

In this section, we further analyze the ability of the model to mitigate the impact of data sparsity and the impact of two hyper-parameters on the model performance.

**Effect of Train Data Ratio in Fine-tuning.** In NLP, the PLMs could alleviate data sparsity problems in the downstream task. In this part, we conduct experiments to analyze whether this phenomenon still exists in expert finding, we adjust the training data ratio in the fine-tuning stage. We employ $[20\%, 40\%, 60\%, 80\%, 100\%]$ all data in **Physics** and **Tex** dataset as training data to fine-tune our model, meanwhile, the ratios of validation data and testing remain the same (i.e., $10\%$) with the main experiments. And we use the recent competitive method PMEF as the baseline method.

The results are shown in Figure 3. As we can see, the two model performance substantially drops when less training data is used. Nevertheless, we can find that there are growing gaps between the results of CPEF and PMEF with the reduction of training data, which indicates the advantage of the pre-trained question representation model is larger when the training data is more scarce. This observation implies that the CPEF shows the potential to alleviate the data sparsity issue via fully mining the knowledge from limited question information.

**Effect of Masking Token Ratio.** In our pre-training, we employ the masked language model to capture the CQA knowledge, and hence the mask ratio is an important hyperparameter of CPEF. We have varied the ratio in $[0.1, 0.15, 0.2, 0.25, 0.3]$ for exploring the model performance w.r.t different mask ratios.

The experimental results are shown in Figure 4. We can find that all metric results increase at first as the masking ratio increases, reach the maximum value (i.e., the best model performance), and then

degrades. When the ratio is small, the BERT model could not capture adequate CQA language knowledge, which could reduce the performance of downstream tasks. On the contrary, when the ratio is large, the [M] symbol appears in pre-training stage more frequently, which could intensify the mismatch between pre-training and fine-tuning. Hence, we set up the ratio of masking tokens to $0.2$ during the pre-training stage.

**Effect of Temperature $\tau$.** In our method, we employ the contrastive task to pre-train question representations, and hence, the temperature $\tau$ plays a critical role during pre-training, which could help mine hard negative samples. Figure 5 shows the curves of model performance w.r.t. $\tau$. We can observe that: (1) Increasing the value of $\tau$ (e.g., $\geq 0.2$) would lead to poorer performance. This is because increasing the $\tau$ would cause the model could not distinguish hard negatives from easy negatives. (2) In contrast, fixing $\tau$ to a too small value (e.g., $\leq 0.05$) would hurt the model performance. The reason may be the gradients of a few negatives dominate the optimization, which would lose balance during model optimization. In a nutshell, we suggest setting the temperature $\tau$ is $0.1$.

## 6 Conclusion

In this paper, we propose a domain contrastive pre-training expert finding model to learn more effective representations of questions and experts. The core of our method is that we design a contrastive framework between question bodies and titles to integrate into the pre-training phrase with the domain CQA corpus. In this way, our method could obtain precise features of questions as well as experts. In addition, the personalized tuning network could further enhance expert representations. Extensive experiments on several CQA datasets validate the effectiveness of our approach, which could outperform many competitive baseline methods.

# 7 Limitations

Our contrastive pre-training method has achieved better performance in expert findings, there are the following limitations that we would like to explore in the future yet. Firstly, although PLMs have been widely used in various fields, the large computation and resource costs are still issues for PLM-based models. Hence, we would consider using further distillation operations for inference efficiency. Secondly, the experiments are conducted in different domains of StackExchange.com, and it would be more convincing if the datasets contains other CQA platforms such as Quora.com or Zhihu.com.

# Acknowledgement

This work was supported by the Shenzhen Sustainable Development Project under Grant (KCXFZ20201221173013036).

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
