# OpenReview forum: "Contrastive Pre-training for Personalized Expert Finding"
_EMNLP/2023/Conference — EMNLP 2023 Findings_

### Official Review · Reviewer_L924 · 2023-08-03

**Typos Grammar Style And Presentation Improvements:** 1) It is suggested to replace "compar…
**Soundness:** 3

**Excitement:**

3: Ambivalent: It has merits (e.g., it reports state-of-the-art results, the idea is nice), but there are key weaknesses (e.g., it describes incremental work), and it can significantly benefit from another round of revision. However, I won't object to accepting it if my co-reviewers champion it.

**Missing References:**

ExpertPLM: Pre-training Expert Representation for Expert Finding (https://aclanthology.org/2022.findings-emnlp.74.pdf)

**Paper Topic And Main Contributions:**

Since existing PLMs are usually  pre-trained on general corpus, directly fine-tuning the existing PLMs with the Expert Finding task may not be the most optimal approach for Community Question Answering (CQA).  Furthermore, such architectures fail to account for the personalized characteristics of experts.
To solve the aforementioned problems, the paper designs a novel expert finding framework. The main contributions include:
1) designing a novel question title-body contrastive pre-training task, which could improve question modeling effectively,
2) adopting a personalized tuning network to learn personalized representations of experts.


**Questions For The Authors:**

A. What exactly does "expert ID" refer to? Is it the username or something else? Furthermore, can the ID information truly represent the personal characteristics of the expert?
B. Why are the lengths of the question body and question title set to 30 and 10 respectively, which are far from the maximum length of Bert(512
characters)? Is it possible to lose information if the length is too short? Is the length setting the same for different datasets?
C. The experimental dataset is different from that mentioned in the comparison method paper, why can't choose the same batch of experimental data?

**Reasons To Accept:**

1) The proposed  title-body contrastive pre-training task  and the personalized tuning network sound reasonable and innovative
2) The performance comparison is well-executed as it involves multiple baselines compared across six datasets, making the results highly convincing.

**Reasons To Reject:**

1) Inadequate method details: The "expert ID embedding" mentioned in Section 4.3 is somewhat confusing as it lacks specific clarification.  It remains unclear whether this ID refers to the registered name of the expert or some other form of identification.  If it simply represents the expert's registered name, its ability to capture the personalized characteristics of the expert is questionable.
2) Insufficient experiments: The "Further Analysis" section of the paper and the experiments of "ExpertPLM: Pre-training Expert Representation for Expert Finding (https://aclanthology.org/2022.findings-emnlp.74.pdf)" share significant similarities. However, there is a lack of experimental analysis specific to certain parameters within the paper itself, such as the length of the question body and title and the number of negative samples (K value).
3) Lack of fair comparison: In Figure 3, the authors compared CPEF with PMEF to demonstrate the advantages of the pre-trained question representation model under data scarcity conditions (line 529-lin534). However, emphasizing the advantages of CPEF through this comparison is unjust since PMEF lacks a pre-training module. To ensure fairness, it is recommended to compare CPEF with another pre-trained model, such as ExpertBert, to showcase the advantage of the innovative pre-training module design of CPEF.


**Reproducibility:**

4: Could mostly reproduce the results, but there may be some variation because of sample variance or minor variations in their interpretation of the protocol or method.

**Reviewer Confidence:**

4: Quite sure. I tried to check the important points carefully. It's unlikely, though conceivable, that I missed something that should affect my ratings.

---

> ### Author Rebuttal · Authors · 2023-08-28
>
> Thanks a lot for all your positive and meaningful comments.
>
> ### R-1 and Q-A (inadequate method details: the "expert ID embedding")
>
> Thanks for your comment.
>
> The expert ID embedding is obtained from the unique ID assigned to each expert. Due to the fact that experts may have different points of interest in the same question, it is unreasonable for different experts to learn the same representation for the same historical question (in the most extreme case, if two experts answer exactly the same question, the model will also learn the same representation for the two experts). Hence, it becomes essential to incorporate personalized information to ensure that the learned vectors are typically specific to each expert. One straightforward and effective approach utilizing the expert's unique ID is to generate a looking-up embedding table similar to word embedding in NLP.
>
> In fact, similar operational methods have been employed in various fields like recommendation systems [1][2] and personalized text generation [3], providing evidence of the effectiveness of utilizing user-unique IDs to achieve more personalized and precise user modeling.
>
> - [1] Wu C, Wu F, An M, et al. NPA: neural news recommendation with personalized attention[C]//Proceedings of the 25th ACM SIGKDD international conference on knowledge discovery & data mining. 2019: 2576-2584.
> - [2] Li L, Zhang Y, Chen L. Personalized prompt learning for explainable recommendation[J]. ACM Transactions on Information Systems, 2023, 41(4): 1-26.
> - [3] Xu H, Liu H, Lv Z, et al. Pre-trained Personalized Review Summarization with Effective Salience Estimation[C]//Findings of the Association for Computational Linguistics: ACL 2023. 2023: 10743-10754.
>
> ### R-2 and Q-B (the length of the question body and title and the number of negative samples)
>
> Thanks for your comment.
>
> This is due to an oversight on our part that has led to your confusion.
>
> * For the length of the question title/body.
>
> In our method, we consider the question title as the primary perspective of the question, and the question body as an augmented perspective. There is a long-tail distribution in the lengths of the question title and body. To maximize coverage of both titles and bodies while minimizing resource usage and training time, we set the title length to 10, which covers 80% of pre-processed titles, and the body length to 30, which covers 75% of pre-processed bodies.
> To enable batch pre-training across various datasets, we standardize the lengths of the title/body in the different datasets.
>
> * For the negative samples.
>
> We selected the same values as the previous paper PMEF [1], i.e., 1 positive sample and 19 negative samples, to ensure the fairness of the experimental comparison.
> - [1] Peng Q, Liu H, Wang Y, et al. Towards a multi-view attentive matching for personalized expert finding[C]//Proceedings of the ACM Web Conference 2022. 2022: 2131-2140.
>
> We will revise the relevant statement in the final version to enhance the clarity of the paper.
>
> ### R-3 (lack of fair comparison: compared with ExpertBERT)
>
> Thanks for your meaningful comment.
>
> The main purpose of this experimental setup is to demonstrate the effectiveness of pre-trained high-quality question representations.
>
> We conduct similar experiments based on ExpertBERT to show the effectiveness of our method.
>
> **Table 1.** Experimental Results of ExpertBERT on the Physics Dataset.
>
> |Training Data Ratio|20%    |40%    |60%    |80%   |100%  |
> |-------------------------|---------|----------|---------|---------|---------|
> |P@3|0.3232|0.3908|0.4668|0.5366|0.5803|
>
> The experimental results fall in between CPEF and PMEF. This is because ExpertBERT is a preliminary pre-trained model based on MLM, whereas CPEF incorporates the question body and employs contrastive learning tasks, which could help the model to learn enhanced question embeddings.
>
> These experimental results will be covered in the final version.
>
> ### Q-C (experimental dataset)
>
> Thanks for your meaningful question.
>
> In our method, a portion of our dataset overlaps with the dataset used in the previous paper (e.g., English, Electronics). StackExchange comprises a wide range of diverse datasets. The datasets compared in different papers lack consistency. We selected a subset of commonly used datasets based on existing papers, which may not encompass all available datasets.
>
> If required, we can conduct additional experiments on relevant datasets.
>
> ### Missing Reference
>
> Thanks for the heads up.
>
> As stated above, ExpertPLM is concerned with expert-level pre-training, whereas this paper is concerned with question-level pre-training.
>
> We've evaluated the ExpertPLM performance and the experimental results are as follows:
>
> **Table 2.** ExpertPLM Experimental Results.
> |           |MRR|P@3|NDCG@10|
> |----------|-------|------|---------------|
> |English|0.4187|0.4833|0.5123|
> |Tex|0.6033|0.7023|0.6711|
> |Electronics|0.5435|0.6305|0.6252|
>
> Based on the results, we observe that ExpertPLM performs worse than CPEF. The possible reasons are as follows:
> 1) Our model encompasses more comprehensive information, including the utilization of question body information, whereas ExpertPLM does not utilize the question body information.
> 2) Precisely modeling questions is the foundation of precisely modeling experts, as expert embedding learning heavily relies on her/his historicalal answered questions.
>
> These experimental results and analyses will be included in the final version.
>
> Thank you for your suggestions regarding the presentation and writing of our paper. We will conduct a thorough proofreading and have invited a native English speaker to assist in polishing the writing.

---

### Official Review · Reviewer_hASg · 2023-08-04

**Soundness:** 3

**Excitement:**

3: Ambivalent: It has merits (e.g., it reports state-of-the-art results, the idea is nice), but there are key weaknesses (e.g., it describes incremental work), and it can significantly benefit from another round of revision. However, I won't object to accepting it if my co-reviewers champion it.

**Missing References:**

Tianyu Gao, Xingcheng Yao, and  Danqi Chen, SimCSE: Simple Contrastive Learning of Sentence Embeddings, EMNLP 2021


**Paper Topic And Main Contributions:**

This paper proposes techniques for improving question representation in the expert finding task through contrastive learning, and for finding experts that match personalized preferences by learning personalized expert representations.

In particular, it suggests viewing the question title and body as a positive pair, offering a solution to overcome the limitations of data augmentation methods, which could potentially be laborious work.
For personalized tuning, they proposed a mechanism grounded in attention-based processing of historically answered questions.

The authors assert the superiority and effectiveness of the proposed method based on the experimental results.

**Questions For The Authors:**

1. It appears that the explanation for the necessity of personalized fine-tuning is insufficient.
2. Shouldn't there be a comparison with existing data augmentation methods, even if it's limited to a small amount of data?
3. It seems necessary to compare the proposed positive pair method with other unsupervised contrastive learning methods, such as SimCSE: Simple Contrastive Learning of Sentence Embeddings (EMNLP 2021).





**Reasons To Accept:**

1. When comparing the experimental results of the proposed method with other baseline methods, the proposed method exhibits superior performance.
2. The authors effectively performed question representation without manual tasks like labeling by introducing contrastive learning techniques to unlabeled data.

**Reasons To Reject:**

1. It appears that the explanation for the necessity of personalized fine-tuning is insufficient.
2. Shouldn't there be a comparison with existing data augmentation methods, even if it's limited to a small amount of data?
3. It seems necessary to compare the proposed positive pair method with other unsupervised contrastive learning methods, such as SimCSE: Simple Contrastive Learning of Sentence Embeddings (EMNLP 2021).





**Reproducibility:**

4: Could mostly reproduce the results, but there may be some variation because of sample variance or minor variations in their interpretation of the protocol or method.

**Reviewer Confidence:**

4: Quite sure. I tried to check the important points carefully. It's unlikely, though conceivable, that I missed something that should affect my ratings.

---

> ### Author Rebuttal · Authors · 2023-08-28
>
> Thanks a lot for all your meaningful comments.
>
> ### R-1 and Q-A (explanation of personalized fine-tuning)
>
> Thanks for your comment.
>
> Given that experts often have distinct perspectives on the same question, it is impractical for different experts to learn identical representations for the same historical question (in the most extreme case, if two experts answer the same questions historically, the model will learn the same representation for the two experts). Furthermore, since our pre-training primarily concentrates on question modeling, it becomes essential to devise a personalized fine-tuning network to ensure that the learned embeddings are typically expert-specific.
>
> ### R-2 and Q-B (comparison with existing data augmentation methods)
>
> Thanks for your comment.
>
> In fact, the data augmentation employed in our method does not involve manual data augmentation, since we can obtain the question titles and bodies naturally from the datasets. Then we take the question body as an augmented view of the question title.
>
> To compare it with different augmentation methods, we **pre-train the model directly based on SimCSE and use SimCSE's data augmentation methods**.
> We select two datasets to conduct the experiment, and the results are as follows,
>
> **Table 1.** Compared with Different Augmentation in SimCSE.
> |                 |MRR|P@3|NDCG@10|
> |--------------|-------|-------|---------------|
> |Electronics|0.5426|0.6303|0.6266|
> |Es|0.5078|0.5658|0.5601|
>
> We can find that the performance is inferior to CPEF. We argue that the reason may be that although such enhancement methods have the potential to learn superior question embeddings, they are still not as effective as directly considering the question body as an enhanced representation of the question title, which is natural to introduce richer information.
>
> These experimental results will be included in the final version of the paper.
>
> ### R-3 and Q-C (compared with SimCSE)
>
> Thanks for your comment.
>
> Our method concentrates more on question modeling while SimCSE focuses more on identifying different sentences. Furthermore, we argue that SimCSE, being a pre-trained model similar to BERT, could serve as the initialization weights for our task.
>
> We **directly employ SimCSE** to evaluate expert finding performance. The experimental results are as follows:
>
> **Table 2.** SimCSE Experimental Results.
> |                 |MRR|P@3|NDCG@10|
> |--------------|-------|-------|---------------|
> |Electronics|0.5316|0.6078|0.5987|
> |Es|0.4701|0.5395|0.5453|
>
> Based on the results, we observe that the experimental results are similar to those obtained with the original BERT (Table 3 in Manuscript). This is attributed to both BERT and SimCSE could be considered as different methods of initialization pre-training weights. However, our method is a combination of question-level pre-training and expert-level personalized fine-tuning. Moreover, as SimCSE specifically emphasizes differentiating embeddings of different sentences during pre-training, it outperforms the original BERT.
>
> These experimental results will be included in the final version of the paper.

---

### Official Review · Reviewer_QLNj · 2023-08-05

**Soundness:** 4

**Excitement:**

4: Strong: This paper deepens the understanding of some phenomenon or lowers the barriers to an existing research direction.

**Paper Topic And Main Contributions:**

In this paper, the authors propose a pre-training framework based on contrastive learning for personalized expert finding in the task of community question answering. Experiments are conducted on several StackExchange datasets. The experimental results show that the proposed method can outperform several conventional baseline methods. The main contribution of this paper is the novel usage of contrastive pre-training tasks for personalize expert finding.

**Questions For The Authors:**

Question A: Instead of BERT, do authors try different pre-trained language models and analyze how the selection of models would affect the performance?

Question B: Could the authors provide some concrete examples about how the proposed method improves the performance?

**Reasons To Accept:**

* Contrastive pre-training is novel for personalized expert finding.
* Extensive experiments with comparisons to many SOTA methods.
* Ablation study shows the effectiveness of each method.

**Reasons To Reject:**

* The technique itself is not novel in NLP while the method is not tailored for the specific research question.
* Lack of concrete examples to demonstrate how the proposed method improves over baseline methods.
* Presentation and writings have room for improvement.

**Reproducibility:**

3: Could reproduce the results with some difficulty. The settings of parameters are underspecified or subjectively determined; the training/evaluation data are not widely available.

**Reviewer Confidence:**

5: Positive that my evaluation is correct. I read the paper very carefully and I am very familiar with related work.

---

> ### Author Rebuttal · Authors · 2023-08-28
>
> Thanks a lot for all your positive and insightful comments.
>
> ### R-1 (the technique itself is not novel in NLP)
>
> Thanks for your comment.
>
> We agree that the contrastive pre-training technology is not novel in NLP. However, there is no effective method to explore the effectiveness of contrastive learning in CQA or expert finding.
> In fact, the expert finding task has the following unique characteristics:
>
> 1) Existing general PLMs are usually pre-trained on the general corpus, such as BookCorpus, Wikipiedia, which have some gaps with the CQA domain and expert finding task. Additionally, we fine-tuned the original BERT model directly on the expert finding task and presented the experimental results in “Table 3”.
> 2) The expert finding task is essentially a recommendation task based on the text modeling. Unlike traditional NLP tasks that solely focus on text modeling, the expert finding task needs to model question and expert simultaneously. Furthermore, different experts have different personalized characteristics. Hence it is non-trivial to directly employ the PLMs in expert finding.
> 3) Unlike certain contrastive pre-training methods in NLP, we do not require the design of additional data augmentation techniques because the question body could serve as a naturally augmented view of the question title.
>
> Consequently, we propose a question-level pre-training and expert-level fine-tuning architecture that enables the learning of more precise question embeddings during pre-training and more personalized expert embeddings during fine-tuning.
>
> We apologize for any confusion caused. In the final version, we will provide a more detailed explanation of the motivation to enhance the clarity and highlight the novelty of our paper for easier comprehension.
>
>
> ### R-2 and Q-B (lack of concrete examples)
>
> Thanks for your comment. We agree that there is indeed a lack of a case study to visualize the effectiveness of our approach more.
>
> Here we present the case study as follows. Given a target question:
> - title: "Magnetic field on the surface of a solenoid",
> - body: "Rotating cylinder behaves like a solenoid, and we can thus compute that the external magnetic field is 0 and that the magnetic field inside it is a constant value  But what is the magnetic field at the surface?".
>
> The CPEF and ExpertBERT recommend different experts for answering the question. Expert A recommended by CPEF is the ground truth, while Expert B recommended by ExpertBERT is false. Partial historical answers for A and B are as follows:
>
> **Table 1.** Expert A.
> |Question|Title|Body|
> |------------|-------|------|
> |1|How can I find the force of a solenoid with a moving plunger|It seems like it should be a simple equation, until I realized that the core isn't magnetized until it is induced, then there is a dipole moment, and then as it moves the core of the solenoid gradually changes from air to the core material|
> |2|Force to extend a solenoid|Suppose a long solenoid has length l, area S and N turns. I want to know the force required to extend it when the current is I.|
> |3|Electric Field in an Infinite Alternating Current Solenoid|I stumbled upon an explanation I can't understand. It goes like this: There's an alternating current solenoid with radius a. We know the magnetic induction is|
>
> **Table 2.** Expert B.
> |Question|Title|
> |------------|-------|
> |1|Magnetic moment precession around magnetic field|
> |2|Moving conducting bar in (changing) magnetic field|
> |3|Condition for the magnetic field|
> |4|induced emf when a wire or coil travel through a magnetic field|
>
> From Tables 1 and 2, we can find Expert B has the potential to answer the target question because he answered "magnetic field" related questions. ExpertBERT recommends Expert B for the target question based on this fact. However, CPEF takes the question body as the augmented view of the question title, which could capture additional rich information for question embedding. In this way, the expert embedding learned by CPEF based on historical questions is more related to the "solenoid magnetic field". Hence, the CPEF model recommends Expert A not B to the target question "Magnetic field on the surface of a solenoid".
>
>
>
> We will include a case study in the final version to better visualize how the proposed method improves the performance.
>
>
>
> ### R-3 (presentation and writings)
>
> Thanks for the suggestions about the presentation and writing in our paper. We will carefully do comprehensive proofreading and invite a native English speaker to help polish the writing.
>
> ### Q-A (different pre-trained language models)
>
> Thanks for your suggestions. We agree that employing a more advanced pre-trained model may improve the effectiveness of the model. However, the main focus of this paper is not the selection of pre-trained models, but rather the designing of the pre-training architecture in expert finding. Therefore, we adopt the widely used BERT as the “base” model.
>
> Moreover, we incorporate SimCSE and RoBERTa as alternative “base” models and subsequently assess their performance. The results are as follows:
>
> **Table 3.** Different Pre-trained Language Models (Physics dataset as an example).
> |              |MRR|P@3|NDCG@10|
> |------------|-------|------|---------------|
> |SimCSE|0.5052|0.5913|0.5750|
> |RoBERTa|0.5049|0.5908|0.5763|
>
> The results indicate that replacing the pre-trained model has minimal influence on the performance since our pre-training strategy is conducting domain-specific pre-training in the CQA field. The existing pre-training model primarily offers improved parameter initialization, which can be viewed as the model "base". Our pre-training corpus and designed tasks would encourage the model to capture more domain-specific information.
>
> In future work, we plan to investigate the application of more advanced pre-trained models for CQA and expert finding.

---

### Meta-Review · Area_Chair_bwJR · 2023-09-22

**Recommendation:** 2

**Metareview:**

This submission studies the Community Question Answering (CQA) problem and adopts the pre-training technique to enhance the representation of questions and answers. In particular, this submission adopts a contrastive learning with some specific design to inject personalized preferences. Nevertheless, the technical novelty is limited and lacks experiments beyond the StackExchange platform. In addition, the performance improvement in many cases is marginal. Besides, the writing of this submission should be improved.

---

### Decision · Program_Chairs · 2023-10-07

**Decision:**

Accept-Findings

**Comment:**

This submission studies the Community Question Answering (CQA) problem and adopts the pre-training technique to enhance the representation of questions and answers. In particular, this submission adopts a contrastive learning with some specific design to inject personalized preferences. Nevertheless, the technical novelty is limited and lacks experiments beyond the StackExchange platform. In addition, the performance improvement in many cases is marginal. Besides, the writing of this submission should be improved.